# Valuing Human Impact of Natural Disasters: A Review of Methods

**DOI:** 10.3390/ijerph191811486

**Published:** 2022-09-13

**Authors:** Aditi Kharb, Sandesh Bhandari, Maria Moitinho de Almeida, Rafael Castro Delgado, Pedro Arcos González, Sandy Tubeuf

**Affiliations:** 1Institute of Health and Society (IRSS), Universite Catholique de Louvain, 1200 Brussels, Belgium; 2Department of Medicine, University of Oviedo, 3204 Oviedo, Spain; 3Institute of Economic and Social Research (IRES/LIDAM), Universite Catholique de Louvain, 1200 Brussels, Belgium

**Keywords:** human impact, lost lives, value of statistical life, natural disasters

## Abstract

This paper provides a comprehensive set of methodologies that have been used in the literature to give a monetary value to the human impact in a natural disaster setting. Four databases were searched for relevant published and gray literature documents with a set of inclusion and exclusion criteria. Twenty-seven studies that quantified the value of a statistical life in a disaster setting or discussed methodologies of estimating value of life were included. Analysis highlighted the complexity and variability of methods and estimations of values of statistical life. No single method to estimate the value of a statistical life is universally agreed upon, although stated preference methods seem to be the preferred approach. The value of one life varies significantly ranging from USD 143,000 to 15 million. While an overwhelming majority of studies concern high-income countries, most disaster casualties are observed in low- and middle-income countries. Data on the human impact of disasters are usually available in disasters databases. However, lost lives are not traditionally translated into monetary terms. Therefore, the full financial cost of disasters has rarely been evaluated. More research is needed to utilize the value of life estimates in order to guide policymakers in preparedness and mitigation policies.

## 1. Introduction

Since 1960, more than 11,000 disasters triggered by natural hazards have been recorded. The number has steadily increased from an annual total of 33 disasters in 1960 to a peak of 441 disasters in 2000 [1]. Hazards such as storms, floods, heatwaves, droughts and wildfires have increased in number, intensity and variability in recent years [2]. Between 2000 and 2019, there were 510,837 deaths and 3.9 billion people affected by 6681 natural disasters [3]. This rising death rate highlights the continued vulnerability of communities to natural hazards, especially in low- and middle-income countries. The Analysis of Emergency Events Database(EM-DAT) shows that, on average, more than three times as many people died per disaster in low-income countries than in high-income nations [1]. A similar pattern was evident when low- and lower-middle-income countries were grouped together and compared to high- and upper-middle-income countries. Taken together, higher-income countries experienced 56% of disasters but lost 32% of lives, while lower-income countries experienced 44% of disasters but suffered 68% of deaths [1].

Disasters datasets usually report the human impact of disasters fairly precisely, and also include the economic impact mainly related to damages to insured goods; for example, EM-DAT, NatCatservice, MunichRe [1,4]. While economic damages of disasters are available in monetary terms, the human impact is measured in different natural units (lost lives, lost life years, disability-adjusted life years (DALY), etc.). Transforming those human impacts into monetary terms is not straightforward. However, it is of great importance in disaster contexts, as it could serve as a vital tool for a multitude of purposes, not limited to informing policy decision making.

Reinsurance companies could utilize this value to generate risk assessments, calibrate loss-estimation models and validate compensation claims; investors and international organizations could make use of it to advise strategic risk mitigation plans; and academic institutions could use it to measure inequalities and identify research gaps. Additionally, for individuals, the perceived disaster severity and knowledge of disaster-related risks might be limited and can be supplemented by providing monetary value to the physical and psychological health risks they might face [5]. Similarly, as the principal focus of health, safety and environmental regulations and many public health-related policies is to enhance individual health, where the most consequential impacts often pertain to reductions in mortality risks, policymakers seeking to assess society’s willingness to pay for expected health improvements need some measures of the associated benefit values to monetize the risk reductions and to facilitate comparison of benefits and costs. In this context, evaluating the global impact of a disaster would rely on using a unique metric to translate both the human and the economic costs of disasters.

Providing a monetary value to lost lives or health losses relies on the value of statistical life literature. The economics and disaster literature today has shown that although it is difficult to ‘put a price on life’, observation of individual and group behaviors seem to indicate otherwise. People regularly weigh risks and make decisions through a cost–benefit analysis framework, where they weigh the willingness to pay for risk reduction and the marginal cost of enhancing safety [6,7]. According to Kniesner and Viscusi (2019) [8], the value of statistical life can be defined as the local trade-off rate between fatality risk and money. The utility associated with reducing a risk must compensate for the disutility associated with the cost of reducing that risk. This argument is further strengthened by the cost assessment of intangible effects of natural disasters in the literature in welfare economics [9,10]. Individuals derive welfare from non-market goods such as environmental and health assets in more ways than only direct consumption [11]. For example, does the cost of reinforcing and strengthening buildings in a seismically active zone and ensure earthquake resistance save enough lives and prevent enough injuries that, in the long run, individual productivity for the state overshoot the costs exhausted by the state [12]?

This review aims to provide an overview of the methodologies used to evaluate the value of life in a natural disaster context and to present the differences in values of statistical life calculated using these alternative methodologies. The review also highlights the areas in the literature where more research is needed. To this end, the first section of this review reports the methodology for the selection and analysis of the literature. The second section explains the results of the analysis. Finally, we discuss the results and shortcomings of the current literature and draw conclusions from the study.

## 2. Methodology

We conducted a review of the literature reporting on the value of life in disasters adhering to the Preferred Reporting Items for Systematic Reviews and Meta-Analyses (PRISMA) guidelines [13]. The research question was formulated with the collaboration of the co-authors and the search strategy was then developed following extensive discussion.

### 2.1. Search Strategy

Several databases were used to search for literature, including PubMed MeSH, EMBASE and ECONLIT. In addition to this, the search was also performed in SCOPUS and Google Scholar so as not to miss any relevant papers, but only the first 200 results sorted by relevance were picked up from the two latter databases. We then screened the references of included full texts to identify any potential misses from our initial search strategy.

Various keywords synonymous to the two concepts “Value of Life” and “Disasters” were identified to undertake the search for literature. For “Value of life”, words and phrases such as cost of life, value of statistical life, VSL, willingness to pay, value of life lost and economic value of life were identified as relevant. Similarly, for “Disasters”, two additional terms, i.e., natural disasters and hazards, were used. The two concepts were searched separately as one and two, and then the combination of one and two was searched to obtain the results. More details about the search strategy are available in Appendix A.

### 2.2. Eligibility Criteria

The primary inclusion criteria for the search were peer-reviewed articles or gray literature such as conference papers, dissertation and discussion papers on disasters and value of life written in English from 2000 to 2020. We included studies that primarily quantified the value of life in a disaster setting and studies discussing methodology of estimating the value of life without providing a value by itself. No geographical limitations were set.

### 2.3. Data Collection and Analysis

The hits from different databases were exported onto Mendeley citation manager (Mendeley version 1.19.8) for subsequent screenings. Duplicates were excluded first. Titles and abstracts were then screened, and finally, full texts were screened for the papers included after abstract screening, excluding papers clearly outside the scope of this study. All uncertainties about eligibility were discussed between three co-authors (SB, MMA, ST) in all steps of the selection process.

Several papers were excluded in subsequent screening steps. Papers only talking about environmental pollution and climate change without a reference to natural disasters were excluded, as these topics are quite broad and, if not a cause for natural disasters, fall outside the scope of this study. Additionally, articles mainly concerned with terrorism, conflicts and landmines were not included in the final selection. Other categories of papers that were excluded were coal mine accidents, traffic accidents and forest fires. Papers solely talking about housing insurance and policy recommendations were also excluded. A total of five papers were requested directly from the authors as they could not be accessed online.

A data extraction form was developed for this review after consultation with the authors. The data extraction form recorded the descriptive aspect of all the studies included in the review, including methodology used to calculate the value of statistical life (VSL), results, strengths and limitations. This form was then pilot tested to ensure all the information was covered. The excluded studies were also tested against the form to check why they did not fit the form and revised as needed in subsequent steps. More details about the form are available in the Appendix A.

We first provided a descriptive overview of the included studies in terms of disaster types, the year in which studies were published, distribution of studies among countries according to the level of income as classified by the World Bank, simple geographical distribution and methodologies mentioned in the studies which were used to calculate the VSL. We then synthesized the information provided according to major predefined themes, such as methods of estimation of VSL, calculated VSL, and variations in VSL by geographical regions. These were identified before the analysis following discussions within the research team. Additionally, the possibility of emerging themes was considered and actively looked for during identification and processing of predefined themes.

## 3. Results

### 3.1. Descriptive Overview of Included Studies

The initial search yielded a total of n = 2121 articles, coming down to n = 2084 after duplicates were removed. After screening titles and abstracts, n = 115 papers were considered for full text screening. Subsequently, a further n = 87 articles were excluded and two additional papers were excluded during the data extraction process. In addition to the remaining n = 26 papers for the review, one article was included from the reference screening, making the final count of papers for the review n = 27. The detailed process of article selection is presented in a PRISMA flow diagram (Figure 1) [13].

The biggest proportion of the included papers (n = 8, 29.6%) focused on value of life lost due to floods. This was closely followed by papers discussing unspecified disasters or disasters in general (n = 5, 18.5%). Five articles (18.5%) focused on earthquakes specifically, followed by three papers (11.1%) examining the value of life in the context of avalanches and rockfalls. Two articles (7.4%) discussed tornadoes and three papers (11.1%) dealt with a group of disasters consisting of four types of disasters, namely flood, drought, alpine and coastal hazards. One article (3.7%) was about heatwaves (Figure 2).

Most studies (n = 16, 59%) concerned countries classified as high-income countries by the World Bank, including four papers (15%) from the United States of America (USA), three (11%) from the Netherlands and two each (7%) from Switzerland and Australia. Germany, Austria, Russia, Italy, New Zealand and Japan also had one article each in the final pool. Four studies (15%) were from upper-middle-income countries, including two studies from China and one each from Russia and Iran. Only one paper considered a lower-middle-income country, namely Vietnam. Four papers (15%) were not specific to any country and discussed the value of statistical life in general, without geographical consideration. Finally, one paper (3.7%) talked about developing countries in general while talking about value of life and reconstruction costs resulting from earthquakes.

Regarding where the articles were published, all but 4 out of 27 articles (85%) were published in peer-reviewed journals. As we included gray literature, two out of the four were discussion papers, one was a conference proceedings and the remaining one was a doctoral dissertation. The included studies were published in a variety of disaster-related, economics, policy and environmental journals.

### 3.2. Methods Used to Estimate Value of Life

A number of methods used to estimate the value of life were highlighted after reviewing the literature. Table 1 summarizes the different methods used in the included literature.

(a)Revealed preference methods.

The revealed preference method utilizes observed behavior among the individuals that has already occurred and makes use of this to approximate suggested willingness to pay for a change in mortality risk. This method has an advantage over the stated preference approach in that if a person pays a certain amount for a commodity, it is known with conviction that the same person’s WTP for that commodity is at least the amount he/she is willing to pay. The four methods used to reveal preferences include: (a) the hedonic pricing method; (b) the travel cost method; (c) the cost of illness approach; (d) the replacement cost method [14,15,16].

(b)Stated preference methods.

In contrast with revealed preference methods, the stated preferences method creates a hypothetical market in a survey. It parallels a market survey and estimates a willingness to pay for hypothetical reduction in mortality risks, since it resembles market behavior. In addition, stated preference methods incorporate both active and passive use of a commodity by the consumer. Direct or active values arise when an individual physically experiences the commodity, while passive or indirect values entail that an individual does not directly experience the commodity. The three methods used for stated preferences include: (a) the contingent valuation method; (b) the choice modeling method; (c) life satisfaction analysis [17,18,19].

(c)Non-behavioral methods

Non-behavioral methods are not necessarily based on human choices and cognitive biases which affect the choices subconsciously. They include the human capital method (HCM) [20] and life quality index method (LQI) [21] to estimate the valuation of statistical life, and they are used to elicit the value of an individual in a society in the absence of a possibility to conduct a survey pre- or post- disaster.

In the selected literature, 7 papers out of 16 used stated preference methods. Within stated preference methods, two papers used choice modeling, while the other five used a contingent valuation method.

Papers using choice modeling method included Bockarjova et al. (2012) [22] and Rheinberger (2011) [23]. While Bockarjova et al. (2012) [22] carried out a choice modeling experiment via an internet-based questionnaire and elicited responses from people living in flood prone areas in the Netherlands in two separate studies, Rheinberger (2011) [23] undertook a choice experiment by recruiting respondents via a phone call prior to a mail survey.

For contingent valuation method, Leiter et al. (2010) [24] used face-to-face interviews and elicited people’s willingness to pay to prevent an increase in the risk of dying in a snow avalanche. Similarly, Hoffmann et al. (2017) [26] used a computerized payment card method to estimate the willingness to pay to reduce mortality risk in Chinese population living in four different cities in China. In contrast, Ozdemir (2011) [25] used a contingent valuation method as well, but used a mail survey to elicit willingness to pay to reduce the risks from tornadoes in the USA.

For non-behavioral methods, Dassanayake et al. (2012) [35] used a quality of life index method to evaluate intangible flood losses and integrate them into a flood risk analysis.

Other papers used one or a combination of methods. For example, Porfiriev (2014) [31] approached the economic valuation of human losses resulting from natural and technological disasters in Russia using the theory of welfare and an international comparative approach. Cropper and Sahin (2009) [12] used the comparative approach, along with transferring the VSL from USA to a whole list of countries classified by income groups by the OECD to estimate VSL.

### 3.3. Values Provided in the Literature

There was a wide range of VSL values in the literature, ranging from ISD 143,000 to 15 million for one life [12,25]. Table 2 summarizes the estimated value of statistical lives in the articles included in the review. Disaster types range from natural disasters to technological disasters with some disaster types appearing more often than others in the literature, with earthquakes and floods being the most common. The VSLs appeared to increase over the years: while it was estimated to be USD 0.81 million in 2005 in Switzerland in the context of avalanches [34], it was evaluated between USD 6.8 and 7.5 million in 2011 [23].

## 4. Discussion

Disasters are complex events, and the assessment of losses they have caused is a compounded task. This review’s exploration of literature estimating the value of statistical life with regard to disasters highlighted the complexity and variability of the estimation of values of statistical life and the methods involved.

The geographical locations of studies included in the review showed the parts of the world where most of the studies were focused. An overwhelming majority of studies estimated the value of statistical life in high-income countries. The main reasons for this are related to the data availability and the investment made by developed countries in research and development for the advancement of science in general [38]. Low- and middle-income countries often experience several disasters occurring year round, and become trapped in a loop of disaster recovery and management annually. Amid ever-present financial constraints, disaster risk reduction and management planning to deal with disasters and their impact in the country therefore becomes much more demanding [39].

The estimation of economic damages due to disaster in a low-resource setting can also be challenging. Not all the houses, agricultural land, crops and other assets are insured in low- and middle-income countries. The insurance coverage is relatively small if not non-existent in these countries [40] and the data to quantify the impacts of disasters, such as the number of deaths, missing, affected population as well as reconstruction costs, are often incomplete and not well recorded. So, the unavailability of appropriate information becomes a big challenge in the first step of conducting research. This might be the reason why low- and middle-income countries are not well represented in studies estimating the value of life in disasters. As a result, the lack of studies in low- and middle-income countries can lead to a certain degree of extrapolation of results found in VSL calculation in high-income-country-based studies.

Furthermore, we note that the majority of articles measuring the value of life were about floods. Floods are indeed the most common type of disasters. In an analysis of disasters recorded in the EM-DAT database from 2000 to 2019, nearly half (n = 3254) of all recorded events (n = 7348) were floods [41]. However, there are many other types of disasters, and it is important to rely on such studies where those disasters were considered when measuring the value of a statistical life.

Methods used for VSL estimations showed significant diversity among the articles included in this review. Although the stated preferences method is the most frequent, it is closely followed by the adaptation method. There could be various reasons for this difference in methodologies across the literature. For instance, non-marketed good with no complementary or substitute market good may not have readily available individual data, and hence may lead the researchers to undertake stated preference methods with which to elicit people’s willingness to pay to reduce a hypothetical disaster risk through surveys [19]. The scope of the study and the budgetary constraints may also explain why a researcher chooses one method over the other. Additionally, the characteristics of the survey participants are another important factor, as they influence the type of survey that can be conducted and the methodology adopted. For example, if the target population is old and poor, face-to-face interviews in respondents’ private homes might be more suitable than internet-based questionnaires [42,43].

There was a wide range of monetary values of the VSL in the literature. These differences could be due to the level of income of the country where the disaster occurred [40]. The method of calculation could be another reason for such differences, for example, as consumers optimize their lifetime utility, thus neglecting intergenerational (long-term) utility, using willingness to pay (WTP) methods for a reduction of risk can often lead to overestimated values [44,45]. It could also simply be due to the differences in cultural norms between countries [40]. Furthermore, the context and the aim of the research and its evolution over the years might also explain variations across the studies. Further studies are required to establish a concrete cause for this observation. It should also be highlighted that low VSL estimates in low-income countries do not inherently mean that a human life is worth less. It could simply reflect individual income, the cost of commodities and the value of currency [8,46].

This study presents a number of limitations. First, the review only included articles published in English, and some studies may exist in other languages. Second, papers that did estimate a VSL considered a range of different methods, and therefore direct comparison of estimated values was not straightforward. Papers referring to economic impact in terms of natural environment or animals were also excluded, as they do not refer to value of statistical life; however, they can be important for calculating overall economic cost of disasters [47,48].

## 5. Conclusions

This study aims to explore literature estimating the value of statistical life with regard to disasters through a systematic review. After applying the inclusion criteria on the 2121 articles found in the initial keywords search, only 27 articles were included for final review. In the included literature, several attempts at estimating the value of statistical lives in disasters were identified; however, there was no consensus on the method used, and few investigations were carried out in a low- and middle-income country context. This review therefore provides a limited view of the value of statistical life calculations in disaster settings, which may become useful when implementing disaster risk reduction policies and calculating global losses incurred due to disasters. It reveals that an agreed, robust and multi-sectoral approach for the disaster and economics community remains to be defined.

## Figures and Tables

**Figure 1 ijerph-19-11486-f001:**
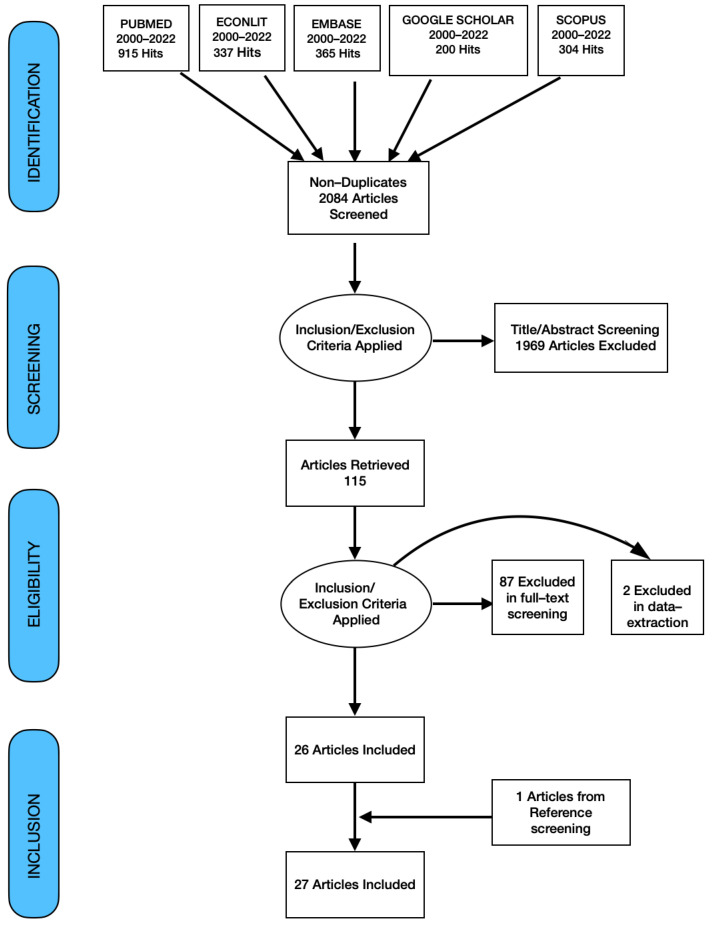
PRISMA flow chart of search, inclusion and exclusion screening and accepted studies of the review. Source: Authors.

**Figure 2 ijerph-19-11486-f002:**
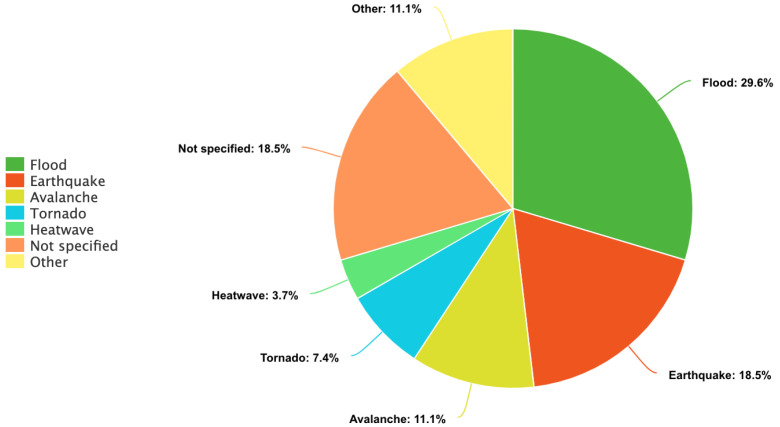
Numbers of included studies by type of disaster. Source: Author.

**Table 1 ijerph-19-11486-t001:** Value of statistical life estimation methods.

Methods	Description	Reference
Stated Preference: - Choice Modeling (CM) - Contingent Valuation Method (CVM)	VSL is based on the willingness to pay for a reduction in the risk of dying. CM differ from CVM in that respondents make repeated choices between different risk attributes.	[22]; [23]; [24]; [25]; [26]; [27]
Adaptation Method	VSL is based on the marginal rate of substitution of disaster loss decrease to disaster prevention investment, which is measured as the ratio of total benefits to total cost.	[28]; [29]; [30]
International comparison method/VSL transfer	VSL is based on available VSLs in other countries. It is converted using the income elasticity of VSL and the Gross Domestic Product per capita. A greater income elasticity of VSL when transferring from a higher- to lower-income country results in lower estimates of the VSL.	[31]; [32]; [12]
Human Capital Method	VSL is based on individual’s future contributions to social production measured by either the gross human capital (foregone earnings) or the net human capital (forgone earnings minus future consumption).	[33]; [34]
Quality of life index method	VSL is measured as an acceptable level of public expenditure to reduce the risk of death that results in improved quality of life.	[35]
Theory of welfare method	VSL is based on lost wellbeing using welfare theory, which considers the value of one’s life at the aggregate economy-wide level, including expected incomes, failed inputs and benefits.	[31]

**Table 2 ijerph-19-11486-t002:** Estimated values of statistical life in included articles.

Reference	VSL (in Millions USD *)	Countries	Disaster Types
Cropper and Sahin (2009) [12]	0.143 (Low-Income-Country) 4.27 (High-Income-Country)	Not Specified	Not Specified
Porfiriev (2014) [31]	0.19 (International comparison) 0.33 (Welfare method)	Russia	Natural and technological
Hoffmann et al. (2017) [26]	0.61	China	Not Specified
Sadeghi et al. (2015) [32]	0.73–1.4	Iran	Earthquakes
Fuchs and Mcalpin (2005) [34]	0.81	Switzerland	Avalanches
Daniell et al. (2015) [33]	2.2	Australia, calculations applied to case studies in Turkey and Croatia	Earthquakes
Cheng (2018) [36]	2.34	Australia	Heatwave
Leiter et al. (2010) [24]	2.3–4	Austria	Avalanches
Dassanayake et al. (2012) [35]	2.5–9.2	Germany	Floods
Zhai et al. (2003) [28]	3.3–9.2	Japan	Floods
Johansson and Kristrom (2015) [29]	5.2–12.8	USA	Floods and storms
Rheinberger (2011) [23]	6.8–7.5	Switzerland	Snow avalanche and rockfalls
Barbier (2022) [27]	1.25–7.7	Italy	Earthquake
Bockarjova et al. (2012) [22]	9.6	The Netherlands	Floods
Hammitt et al. (2019) [30]	10	China	Not specified
Ozdemir (2011) [25]	15	USA	Tornado

* Values were converted into United States Dollars (USD) in respective years. Source: Authors [37].

## Data Availability

Not applicable.

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
