# Peer review of "Valuing Human Impact of Natural Disasters: A Review of Methods"

_ijerph, 2022, doi:10.3390/ijerph191811486_

Round 1
Reviewer 1 Report (Previous Reviewer 1)
The paper's subject is interesting but quite limited in terms of sample size and papers exclusion criteria. In fact, for a literature review, a sample of 27 papers is clearly insufficient to reach credible conclusions. The conclusions should highlight the limitations and can’t do generalizations.
The data concerns climate change (lines 4-6) and deaths related to climate disasters (lines 7-8) is presented as a motivation to the paper approach/subject. Contrary to the author's response statement, I don’t suggest the extending the search to topics like pollution or climate change (see my first comments/review). I just appeal to a justification/reason for excluding papers on environmental pollution, climate change, forest fires, etc. Although the improved justification/reason for excluding papers that talk about environmental pollution and climate change, this exclusion isn’t coherent with the data relationship pointed out at the beginning of the paper (lines 4-8).
Other details to improve:
The Abstract and Keywords should be included in the paper.
Figure 2: Please move this figure to line 161.
Lines 342: Please replace “This review provides an overview of value…” by “This review provides a limited view of value…”.
Author Response
see attached

Reviewer 2 Report (Previous Reviewer 2)
This version has been improved a lot. I think this paper is now acceptable.
Author Response
thank you
Reviewer 3 Report (Previous Reviewer 3)
The reviewer thanks the authors for the reply letter and the updated manuscript. Remarks are correctly addressed and there are no further comments.
Author Response
Thanks
This manuscript is a resubmission of an earlier submission. The following is a list of the peer review reports and author responses from that submission.
Round 1
Reviewer 1 Report
The paper's subject is interesting but quite limited in terms of sample size and papers exclusion criteria. In fact, for a literature review, a sample of 24 papers is clearly insufficient to reach credible conclusions.
A justification/reason for excluding papers on environmental pollution, climate change, forest fires, etc. is missing.
In my opinion, the authors should deepen the work and resubmit it.
Other details to improve:
The Abstract should be included in the paper.
Line 143: Authors should be accurate, e.g. 29% is not the majority.
Lines 143-150: Please replace this paragraph by a graphic.
Table 2: Please put the conversion site as a reference.
Reviewer 2 Report
Overall, this paper should be an interesting and attractive one, but this paper is too simple in its current version. I hope authors could improve the analysis and discussion. Moreover, please also include the appendix at the end of the manuscript.
Line 8-9, I believe this number has underestimated the results. Moreover, please update the results the latest year. Please refer: He, B. J., Wang, J., Zhu, J., & Qi, J. (2022). Beating the urban heat: Situation, background, impacts and the way forward in China. Renewable and Sustainable Energy Reviews, 161, 112350.
Line 11, EM-DAt, full name?
Line 17-19. The most recent data has also reported people may be reluctant to visit hospitals. See: Will individuals visit hospitals when suffering heat-related illnesses? Yes, but…. Building and Environment, 208, 108587.
Section 2.1, I not sure if disasters are enough for literature query since some weather-related disasters especially on extreme heat have been well described a disaster/harzard. Furthermore, I am not sure if the value of life is enough. I cannot get access to the appendix A so that I cannot justify if such two concepts are enough. You may find some literature in the willingness to pay for heat-resilient infrastructure as well. A framework for addressing urban heat challenges and associated adaptive behavior by the public and the issue of willingness to pay for heat resilient infrastructure in Chongqing, China. Sustainable Cities and Society, 75, 103361.
Following the last question, authors have to well define the research scope of this paper.
The analysis should also indicate the components considered in the economic loss. You may find: https://www.melbourne.vic.gov.au/sitecollectiondocuments/eco-assessment-of-urban-heat-island-effect.pdf
Discussion, what kinds of implications for insurance companies?
Reviewer 3 Report
The paper by Sandesh Bhandari, Aditi Kharb, Maria Rodrigues Leal Moitinho De Almeida, Rafael Castro Castro Delagado, Pedro Arcos Gonzales and Tubeuf Sandypresents presents a literature review of methods used to value human lives in a context of natural disasters. After filtering 1640 literature entries in the period 2000-2020, 24 articles are retained and the main methodologies and VSL presented.
The reviewer thanks the authors for the review in an interesting topic. The description of the methodology gives reasonable confidence about the completeness of the included articles. Nevertheless, authors do not clearly identify the gap in knowledge in the studied subject. For example, in the calculation of VSL in a broader scope (not only applied to natural disasters), it is the subject of current discussion whether using VSL based on the willingness to pay (WTP) for a reduction of risk overestimates its value. In that case, VSL overestimation is attributed to the fact that consumers optimize they lifetime utility thus neglecting intergenerational (long term) utility which makes marginal WTP inefficient [1,2]. Could such discussion be extended to the case of natural disasters? Additional remarks are given below:
1. In line 29 it is stated that the monetary value of human impact could also be used by academic institutions, for measuring inequalities and identifying research gaps. Can you comment on that in the reply? No need to modify the document.
2. Why is the search limited to 2020 and not to the present date?
3. The methods listed in table 1 need to be described in more depth to make the review self-contained. The existing description in the second column can be left as it is or synthetized a bit more using keywords, and then each method described in the main text.
Specific comments
EM-DAT needs to be defined.
Statement in lines 40-42 needs supporting references.
Lines 49-50: reference needed.
Consider adding charts or tables presenting data in lines 143-150 and 152-162.
Table 1: DCE not defined before.
References [3] and [28] in the manuscript are the same.
References
[1] Broughel, James and W. Kip Viscusi. 2021. “The Mortality Cost of Expenditures.” Contemporary Economic Policy 39(1): 156–67.
[2] Broughel, James and Dustin Chambers. 2022. Federal Regulation and Mortality in the 50 States. Risk Analysis 42(3): 592-613.
Reviewer 4 Report
Comments:
Abstract
1. “The value of one life varies significantly ranging from 143,000 to 15 million USD” must be removed. The numbers you used require a reference that is not advisable to be included in the Abstract.
Introduction
2. Information about disasters caused by natural hazards must be updated (line 4, 9); thus, you should also cover the period after 2000 and 2013.
3. Lines 17-18: „Disasters datasets usually report 18 fairly precisely the human impact of disasters”. Please, include several examples of such datasets.
4. Line 2-38: Split the text into several paragraphs.
5. Line 40-42: „The economics and disaster literature today has shown that although it is difficult to ‘put a price on life’, observation of individual and group behaviors seem to indicate otherwise.” Include some references for the mentioned ec. and disaster literature.
Methodology
6. Why did you consider only the first 200 results sorted by relevance for Scopus and Google Scholar? How many search results did you consider for PubMed MeSH, EMBASE, ECONLIT?
7. Why did you consider grey literature for the review, knowing that the information provided is not always reliable?
8. Fig. 1: What does the number of citations mentioned for each database refer to?
9. Lines 205-206: Citation is needed for the„There was a wide range of VSL values in the literature, ranging from 143,000 206 to 15 million US dollars for one life.”
Discussion
10. The Discussion section answers the objectives of the review. However, I suggest taking several paragraphs from this section and creating a new one: Conclusions.
